# TSLM: Tree-Structured Language Modeling for Divergent Thinking

**Doyoung Kim**[12*] **Jaehyeok Doo**[1*] **Minjoon Seo**[1]

[1]KAIST AI, [2]NYU
doyoungkim@nyu.edu

## Abstract

Language models generate reasoning sequentially, preventing them from decoupling irrelevant exploration paths during search. We introduce Tree-Structured Language Modeling (TSLM), which uses special tokens to encode branching structure, enabling models to generate and selectively expand multiple search paths within a single generation process. By training on complete search trees including both successful and failed attempts, TSLM learns to internalize systematic exploration without redundant recomputation of shared prefixes. TSLM achieves robust performance and superior inference efficiency by avoiding the multiple independent forward passes required by external search methods. These results suggest a new paradigm of inference-time scaling for robust reasoning, demonstrating that supervised learning on complete tree-structured traces provides an efficient alternative for developing systematic exploration capabilities in language models.

## 1 Introduction

Complex reasoning often requires exploring multiple solution paths before converging on an answer. Consider solving the Game of 24 with numbers [8, 4, 3, 6]: a systematic approach would branch out $8+4$, $8-4$ and other combinations simultaneously, rather than committing to a single path early. However, current language models generate solutions sequentially, making it difficult to systematically explore alternatives or recover from early mistakes.

Recent reasoning models like o1 OpenAI (2024) and DeepSeek-R1 DeepSeek-AI et al. (2025) have shown impressive capabilities through extended reasoning traces, but still fundamentally operate as sequential generators. While they may internally consider multiple options, they cannot explicitly decouple parallel branches within their generation process. External methods like Tree-of-Thought Yao et al. (2023) address this through post-hoc search, but require multiple independent model calls and external orchestration.

The key challenge is that models cannot coherently construct diverse exploration branches. Sequential models collapse all reasoning into one path. External search methods like Tree-of-Thought sample multiple trajectories independently, but this produces fragmented distribution where search

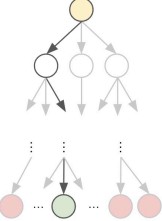

(a) **Sequential thinking:** A purely sequential approach that traverses one deterministic path.

(b) **Tree-structured thinking:** A branching exploration that expands multiple possibilities simultaneously.

Figure 1: **Sequential vs. Tree-Structured Reasoning. (a)** Sequential approaches commit to single paths, limiting exploration of alternatives. **(b)** Tree-structured approaches systematically explore multiple possibilities, enabling recovery from mistakes and comprehensive solution space coverage. TSLM bridges this gap by teaching language models to generate tree-structured explorations natively.

---

* Denotes equal contribution.

sample is drawn without coordination, leading to redundant overlap in some regions while missing critical branches in others. The model has no mechanism to systematically reconstruct the complete search space.

TSLM addresses this through *coherent tree generation*: instead of sampling multiple independent trajectories, the model generates the complete branching structure in one forward pass using special tokens (`[SEP]`, `[FAIL]`, `[GOAL]`) to mark viable paths, dead ends, and goals. During training, models learn to systematically construct diverse branches for each node. During inference, the model generates this coherent tree structure, then selectively "stitches" each branch into isolated contexts for expansion, maintaining the complete search topology rather than hoping parallel samples will cover it.

This approach enables models to learn the complete reasoning process by including both successful paths and failed attempts, leading to more robust problem-solving capabilities. Our experimental results demonstrate that this internalized search strategy not only outperforms sequential modeling baselines but also exhibits remarkable parameter efficiency and emergent capabilities that arise naturally from learning complete exploration patterns rather than just final answers.

## 1.1 OUR CONTRIBUTIONS

This paper makes the following contributions:

- We introduce *Tree-Structured Language Modeling* (TSLM), a token-based serialization framework that enables sequential language model to learn tree-structured reasoning through supervised learning on cohesive search traces

- TSLM demonstrate consistent performance gains across diverse tasks: 100% vs. 17% on Game of 24, robust extrapolation to unseen, larger environments (91.5% vs. 42.7% for Tree-of-Thought in $20 \times 20$ Gridworld environments), and enhanced performance on open-ended reasoning

- We uncover emergent capabilities including systematic identification of unsolvable problems and robust extrapolation beyond training complexity

- We demonstrate a new test-time scaling paradigm: rather than parallel sampling k independent trajectories, TSLM scales by exploring k branches within a single coherent tree, showing better scaling efficiency and higher convergence accuracy

Our findings challenge the granted assumption that we need inference-time scaling (either via reinforcement learning or scaffolding) is necessary for developing robust reasoning capabilities in language models. Instead, we demonstrate that properly structured supervised learning of tree-formatted reasoning traces may provide a more direct and efficient path toward enhanced reasoning and planning. We provide detailed answers to common questions in Appendix §A. We also release our implementation here.

## 2 BACKGROUND

Contemporary language models generate tokens sequentially, modeling $p(y \mid x) = \prod_{t=1}^{|y|} p(y_t \mid x, y_{<t})$ Brown et al. (2020). For problems requiring exploration of multiple solution paths, this sequential approach has limitations: (1) linear commitment to single paths, (2) error propagation, (3) redundant computation when multiple solutions are needed, and (4) inability to systematically explore alternatives in parallel.

Current multi-path reasoning approaches rely on inference-time scaling methods, either for post-hoc sampling or policy gradient update from self rollouts. Tree-of-Thoughts Yao et al. (2023) samples multiple candidates at each step using external search algorithms, but faces exponential computational costs. Reasoning traces generated from autoregressive models like o1 OpenAI (2024) are due to the number of rollouts for gradient update, but remain constrained by linear generation and may produce redundant information Chen et al. (2025). We provide a more extensive related works in Appendix §B.

## 3 Tree-Structured Language Modeling (TSLM)

We introduce Tree-Structured Language Modeling (TSLM) as a framework to natively incorporate divergent exploration for natural language generation. TSLM differs from standard sequential language modeling by generating multiple possible next actions or statements and linking them into a coherent tree structure.

### 3.1 Modeling Multiple Next Actions

Let $s$ be the current state or partial solution. In a sequential language model, we predict a single next action $a$ from $s$ and transition to $s' = T(s, a)$. By contrast, in TSLM, we represent multiple possible successors:

$$\pi_\theta(s) = \{T(s, a_1), \cdots, T(s, a_k)\},$$

where each $a_i$ denotes a distinct branch and $k$ is the branching factor. TSLM learns to expand $s$ into these $k$ successors within a single forward pass, retaining the relationships among them rather than generating them independently. This branching representation is fundamentally different from sampling k independent trajectories. Each $a_i$ is generated conditionally on the previous actions $a_1, \cdots, a_{i-1}$ at the same node, ensuring systematic coverage of the action space rather than redundant sampling from a marginalized distribution.

### 3.2 Encoding and Decoding with Tree Structure

To enable standard transformer architectures to learn tree-structured reasoning, we develop a serialization scheme that encodes complete search trees into linear sequences. This approach allows us to train language models on tree data while preserving the branching structure.

**Token-Based Tree Serialization.** We introduce special tokens that TSLM learns to encode tree structure:

- `[SEP]`: Indicates a viable action that can be further expanded
- `[FAIL]`: Indicates a non-viable action (dead end)
- `[GOAL]`: Marks the desired goal state
- `[BOS]` and `[EOS]`: Mark sequence boundaries of child expansions for each node

This serialization captures both successful paths and unsuccessful explorations, teaching the model the complete search process rather than just final answers. A detailed worked example showing the complete serialization format is provided in Appendix §C.

**Training Procedure.** During training, we apply standard language modeling loss to the entire serialized sequence:

$$\mathcal{L} = -\sum_{t=1}^{T} \log p(y_t | y_{<t}, x) \tag{1}$$

where $y_t$ includes both reasoning content and structural tokens. This standard cross entropy loss has a non standard training signal. Unlike sequential modeling where each token depends only on its prefix, TSLM tokens depend on tree structure. For example, `[SEP]` after action $a_i$ is conditioned on whether $T(s, a_i)$ is expandable **and** whether it differs from $a_1, \cdots, a_{i-1}$. This structural conditioning teaches the model to:

1. Generate multiple actions at each decision point
2. Assign appropriate viability markers (`[SEP]`, `[FAIL]`, or `[GOAL]`)
3. Structure the exploration systematically

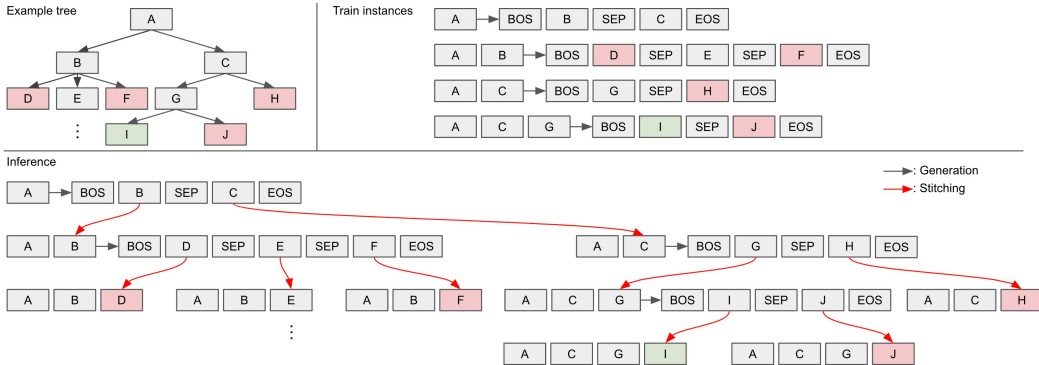

Figure 2: TSLM (Tree-Structured Language Model) is a language model designed for hierarchical exploration in sequence generation tasks. Beginning with an initial state (e.g., "A") and progressing toward a goal state (e.g., "I"), TSLM constructs a tree structure where nodes represent states and branches signify possible paths. During training, the model serializes the tree into linear sequences using special tokens to separate branches and mark the start/end of a sequence, allowing it to learn structured expansions effectively. During inference, TSLM generates multiple branching actions to explore diverse sequences (e.g., expanding from "C" to "G" and "H"). These branches are independently expanded in parallel using a stitching process, enabling broad exploration toward the goal while efficiently pruning unwanted paths.

**Inference Procedure.** During inference, TSLM reconstructs the tree structure:

1. Generate the next reasoning step with multiple candidate actions
2. Parse structural tokens to identify viable branches (`[SEP]`)
3. Fork different branches into independent sequences for parallel expansion
4. Recursively expand each viable state until finding a solution (`[GOAL]`) or exhausting options

This approach enables systematic exploration. While TSLM forks divergent branches into different sequences (involving multiple calls), the construction of the whole search tree is coherent and internalized, avoiding the redundant sampling inherent in Tree-of-Thought.

## 3.3 FORMAL TRAINING OBJECTIVE

For a search tree $\mathcal{T}$ with nodes $\mathcal{N}$, let $\pi(s_i)$ denote the path from root to node $s_i$. Each node $s_i$ generates a token sequence $y_{s_i} = [a_i, m_i]$ where $a_i$ is the action description and $m_i \in \{[\texttt{SEP}], [\texttt{FAIL}], [\texttt{GOAL}]\}$ is the structural marker.

The context for node $s_i$ consists of:

$$\text{ctx}(s_i) = \bigcup_{s_j \in \pi(s_i)} y_{s_j} \cup \bigcup_{\substack{s_k \in \text{siblings}(s_i) \\ k < i}} y_{s_k} \tag{2}$$

where $\text{siblings}(s_i)$ are nodes sharing the same parent, and $k < i$ indicates siblings generated before $s_i$ (enforcing left-to-right ordering within each branching factor).

The training objective becomes:

$$\mathcal{L} = -\frac{1}{|\mathcal{N}|} \sum_{s_i \in \mathcal{N}} \sum_{t=1}^{|y_{s_i}|} \log p(y_{s_i,t} \mid y_{s_i,<t}, \text{ctx}(s_i)) \tag{3}$$

This formulation makes explicit that each node conditions on (1) its ancestral path and (2) previously generated siblings, but **not** on unrelated subtrees. This selective conditioning is what enables context decoupling.

In practice, we serialize the tree using depth-first traversal, converting it into a linear sequence that transformers can process. Crucially, while the input is linear, the conditional dependencies are dictated by tree topology: node $s_i$ depends only on $\mathrm{ctx}(s_i)$ (its ancestors and prior siblings), not on all tokens that precede it in the serialized sequence. This structured conditioning is what distinguishes TSLM from simply concatenating all exploration paths sequentially.

## 4 SEARCH TREE SUPERVISION

### 4.1 TRAINING ON STRUCTURED TASKS

For structured tasks with predefined search trees, we can directly train TSLM to learn and reproduce the tree structure. Formally, let $t \in \mathcal{T}$ be a task with a solution of a finite depth, $T(s, a)$ be a transition function that maps state-action pairs to new states, and $A(s)$ be a finite action space that defines valid actions at state $s$. Since these components are explicitly defined, we can generate the complete search tree and employ TSLM to predict the branching structure $(T(s, a) \mid a \in A(s))$ at each state in the search tree. By directly imitating the predefined tree expansions, TSLM guides the model to faithfully reproduce structured exploration patterns. Examples include board games and planning problems with well-defined rules.

### 4.2 SEARCH TREE SUPERVISION FOR OPEN-ENDED REASONING TASKS

While structured tasks have predefined search trees that TSLM can directly learn from, most real-world tasks lack explicit tree structures, providing only correct answers or gold trajectories. In this paper, we adopt a simple bootstrapping method to construct synthetic training trees by combining model-generated explorations with known solutions; For each training instance, we deploy the Tree-of-Thoughts sampling Yao et al. (2023) from supervision language model to generate pseudo search tree. The process involves:

1. Sampling a set of candidate actions at each state using beam search
2. Building a tree structure by propagating these actions forward
3. Incorporating known gold trajectories as high-priority branches
4. Ordering remaining branches using a reward function $R(s, a)$
5. Deduplicating redundant paths while preserving the tree structure

Algorithm 1 details this procedure. Our approach ensures each training tree contains at least one valid solution while exploring diverse alternatives. The reward-based ordering helps prioritize promising actions, while deduplication prevents redundant search. We adopt the original RAP Hao et al. (2023) reward function to refine exploration by prioritizing promising branches.[1]

## 5 EXPERIMENTAL RESULTS

### 5.1 BASELINES FOR COMPARISON

**Baseline Model Architecture** We compare TSLM to the following baselines:

- **Sequence Cloning (SC)**: A standard sequential modeling that clones a single linear sequence of gold Chain-of-Thought (language modeling similar to GPT-3 (Brown et al., 2020) or Llama 3 (Touvron et al., 2023)).

- **Procedure Cloning (PC)**: A sequential modeling that clones Chain-of-Thought reasonings of the entire search trace in a single linear sequence (o1-like reasoning models, trained with systematic supervision(Kim et al., 2024; Yang et al., 2022)).

---

[1]Caveat here is that this is *not* the best method to train TSLM. For example, we can use supervision from better reasoning language models using techniques like ReJump (Zeng et al., 2025). Also we can use self-training with external search methods like SoS (Gandhi et al., 2024). We rather focus on the concept of TSLM itself rather than high-level training techniques.

---

**Algorithm 1** Guided Search Tree Bootstrapping

---

**Input:** Task $\mathcal{T}$, transition $T$, reward $R$, branch factor $k$, supervision model $\pi_\theta$
`Data` $= []$
**for** $t \in \mathcal{T}$ **do**
   Initialize `queue` $= [s]$.
   Gold trajectory $s_0 = s, s_1, \ldots, s_n = g$
   **while** $g \notin$ `queue` **do**
     $tmp =$ `queue`$.pop(0)$
     **for** $i = 1$ **to** $k$ **do**
       `tmpqueue` $= []$
       **if** $i = 1$ **and** $tmp \in \{s_0, \ldots, s_n\}$ **then**
         $a_i = a^*(tmp)$ {Add gold action}
         `tmpqueue`$.add(T(tmp, a_i))$
       **else**
         $a_i \sim \pi_\theta(tmp)$
         **if** $a_i \notin \{a_1, \ldots, a_{i-1}\}$ **then**
           `tmpqueue`.add$(T(tmp, a_i))$
           {Deduplication}
         **end if**
       **end if**
     **end for**
     `tmpqueue` $= \sigma_R($`tmpqueue`$)${Sort by reward}
     `queue`$+ =$ `tmpqueue`
   **end while**
   `Data`.append(`queue`)
**end for**

---

- **GRPO**: A reinforcement learning approach to incentivize reasoning trace during post-training (o1-like reasoning models, trained with GRPO (Shao et al., 2024) objectives, group size = 8)

- **Tree-of-Thought (ToT)**: Scaling the number of inference of **SC** model during test-time using external search algorithms (Yao et al., 2023) with beam search across multiple reasoning paths.

To ensure a rigorous comparison, we evaluate ToT using a **pass@100** metric (success if any of the first 100 terminal nodes is correct), while all other methods (SC, PC, GRPO, TSLM) are evaluated using **pass@1** (single attempt). We chose ToT@100 because it represents the convergence point of the search tree in our experiments, effectively serving as an upper bound for sampling-based scaffolding methods. Thus, comparing TSLM (pass@1) against ToT (pass@100) provides a highly conservative estimate of TSLM's relative performance.

We test models using greedy decoding for sequential methods, and breadth of 5, temperature of 0.3 across ToT experiments. For TSLM inference, we explore different tree traversal and solution selection strategies. Unless otherwise specified, we use Breadth-First Search (BFS) as the default algorithm to systematically explore the generated tree structure until finding a successful solution. (We analyze implications of BFS versus alternative search strategies such as Depth-First Search (DFS) in Appendix §E.1.) For each expansion during the inference, we select the first $k = 5$ actions generated and deduplicate them with exact matching.

To evaluate Tree-Structured Language Modeling (TSLM), we conduct experiments on both structured and unstructured tasks. Our experiments use Llama-3-8B Grattafiori et al. (2024) as base experiments unless specified, comparing TSLM against sequential language modeling baselines. Also, We aim to compare architectural differences rather than scaling effects, using modest training data (less than 10K instances per task) for post-training. We have two task scenarios: Structured Planning Tasks, which are tasks with predefined search trees, while Open-ended Reasoning Tasks are tasks with sundefined solution spaces. Structured planning tasks include Game of 24 and Textualized Gridworld, while Open-ended reasoning tasks include ProntoQA and GSM8K. Refer to Appendix §F for more details regarding each task and supervision examples.

| | Task | pass@1 | | | | pass@100 |
| | | SC | PC | GRPO | TSLM | ToT |
|---|---|---|---|---|---|---|
| §F.2 | Game of 24 | 17.0% | 47.0% | 15.0% | **100%** | 32.0% |
| | Gridworld (i.d) | 78.2% | 99.7% | 24.0% | **100%** | 95.0% |
| | Gridworld (o.o.d) | 33.0% | 81.1% | 6.0% | **91.5%** | 42.7% |
| §F.3 | ProntoQA | 99.7% | 97.5% | 99.8% | **100%** | 100% |
| | GSM8K | 55.8% | 55.9% | 60.8% | **61.6%** | **62.3%** |

Table 1: Success rates across different tasks and methods. Note that ToT results are reported as pass@100 (success if any of 100 attempts is correct), while all other methods are pass@1. Despite this disadvantage, TSLM matches or outperforms ToT on most tasks. The Gridworld results show both in-domain (10×10) and scaling (20×20) performance, highlighting TSLM's robustness when complexity scales beyond training boundaries.

## 5.2 BASE RESULTS

Table 1 summarizes our experimental findings across all tasks, revealing key insights about different reasoning approaches. TSLM consistently outperforms sequential models, achieving perfect accuracy on structured tasks. Most notably, the Gridworld scaling results reveal a striking limitation of Tree-of-Thought: while ToT achieves excellent in-domain performance (95.0% on 10×10 grids), it suffers catastrophic degradation when complexity scales (dropping to 42.7% on 20×20 grids).

This ToT scaling failure is particularly surprising given its strong in-domain performance and sophisticated external search mechanisms. In contrast, TSLM maintains robust performance across the complexity boundary ($100\% \rightarrow 91.5\%$), demonstrating that internalized search procedures generalize better than external search algorithms. For open-ended tasks, ToT shows comparable performance on GSM8K (62.3% vs 61.6%) but this advantage disappears when systematic exploration is needed, as evidenced by Game of 24 results where ToT performs no better than basic sequential methods (17.0%).

## 5.3 FAIR COMPARISON: TEST-TIME SCALING

A critical distinction: TSLM introduces a fundamentally different test-time scaling paradigm. Traditional methods scale by generating $k$ **independent** trajectories through parallel sampling (ToT) or sequential enumeration. Each trajectory recomputes shared prefixes redundantly. TSLM scales by exploring $k$ candidates within a **single, coherent search tree** generated by the model in one forward pass. The model explicitly constructs the branching structure, then we select which branches to verify.

We define test-time scaling for each method:

- **TSLM**: Generates one coherent tree structure, then verifies the first $k$ terminal states (ending with either [FAIL] or [GOAL]) from Breadth-First Search traversal. Crucially, the $k$ candidates come from a single tree construction, not $k$ independent samplings.
- **Procedure Cloning (PC)**: Verifies the first $k$ terminal states from the sequential search trace.

To rigorously compare the effectiveness of internalized search (TSLM) versus external scaffolding (ToT), we evaluate performance under matched candidate budgets. This comparison is crucial because TSLM's "test-time scaling" differs fundamentally from standard parallel sampling. In ToT or standard scaffolding, scaling $k$ implies generating $k$ independent, redundant traces. In TSLM, scaling $k$ means exploring more branches within a *single, cohesive search tree* structure generated by the model. This internalized scaling allows for more efficient resource allocation, as the model can prioritize promising branches without regenerating common prefixes or exploring completely disjoint paths. Figure 3a presents the results on GSM8K as we scale the number of candidates $k$.

TSLM consistently outperforms ToT and PC across all budgets. Notably, TSLM with just a single candidate (61.3%) nearly matches ToT's converged performance (62.3%), and TSLM saturates at a

significantly higher accuracy (67.2%). This confirms that TSLM's training objective successfully internalizes a superior search tree compared to the parallel sampling in ToT.

## 5.4 COMPUTATIONAL EFFICIENCY ANALYSIS

An immediate followup advantage of TSLM is its inference efficiency. Unlike external scaffolding methods that require multiple independent model calls, TSLM generates the complete search tree in a single forward pass (or a few passes if using beam search decoding), sharing computation for common prefixes.

We compared the average wall-clock time required to reach solution convergence across methods. As shown in Figure 3b, TSLM is significantly faster than ToT. ToT suffers from time explosion due to redundant sampling and lack of computation sharing. TSLM is also faster than Procedure Cloning (PC) because TSLM's tree structure allows the model to ignore irrelevant subtrees during inference, whereas PC must process the entire linearized sequence of the search trace.

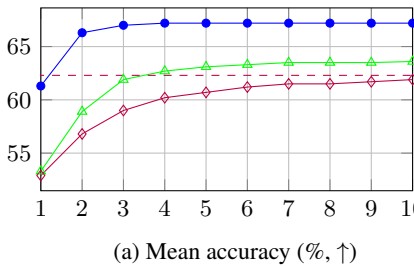
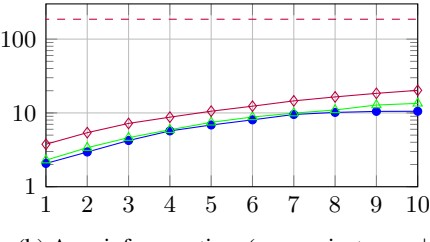

(a) Mean accuracy (%, ↑)

(b) Avg. inference time (sec per instance, ↓)

Figure 3: Comparison of (a) mean accuracy and (b) average inference time when scaling the number of candidates across different methods on GSM8K. Methods: TSLM (●), PC (△), and SC (◇, convergence at ToT@100: - - -). All the cache control during the inference time is done by vLLM (Kwon et al., 2023) v.0.6.6.

## 6 ANALYSIS

Beyond core performance gains, TSLM exhibits several remarkable capabilities that distinguish it from traditional approaches.

## 6.1 IDENTIFYING UNSOLVABLE CASES

One challenging aspect for language models is avoiding hallucination on problems that have no valid solution. Sequential models trained on SC, PC, or GRPO implicitly learn to generate answers within their training distribution, which can be problematic when faced with unsolvable cases. For example, in the Game of 24 task, the numbers 1, 1, 2, and 3 cannot generate 24 through any sequence of arithmetic operations. Since sequential models have not been trained on examples with no solution, they tend to hallucinate and generate invalid answers.

TSLM, however, demonstrates a unique capability to identify unsolvable cases. Quantitatively, across 100 unsolvable Game of 24 instances, TSLM correctly identified 97 cases by terminating without a solution. In contrast, baseline methods (SC, PC, GRPO) failed to identify any unsolvable cases, instead hallucinating invalid solutions (There are 3 unintentional instances where PC correctly refused, only because the model generated excessively long traces that failed to terminate properly due to skewed search tree generation). This suggests the tree-structured exploration enables TSLM to systematically explore the full solution space and recognize when no valid path exists. Unlike sequential models that are pressured to always generate some answer, TSLM's cohesive search tree allows it to confidently determine and declare when a problem is unsolvable.

We position this capability as addressing a fundamental limitation of sequential language models. Hallucination remains a major problem in language modeling, with theoretical work suggesting it's

inevitable in the next-token prediction paradigm (Xu et al., 2025) Frontier labs like OpenAI attempt to address this through post-training to modify refusal behavior (Kalai et al., 2025), but the mechanisms remain unclear and heavily rely on RL approaches.

| SC | PC | TSLM (ours) |
|---|---|---|
|  |  |  |
| 33.0% | 81.1% | **91.5%** |

Table 2: Qualitative comparison of model extrapolation capabilities on Gridworlds of varying sizes (maximum $20 \times 20$). Each heatmap shows performance at different grid dimensions (x,y), with darker colors indicating better performance. The red box indicates the boundary of training data. Overall accuracy shown above each plot.

## 6.2 QUALITATIVE ANLYSIS ON EXTRAPOLATION BEYOND TRAINING DATA

We qualitatively show how TSLM demonstrates strong extrapolation capabilities when tested on larger grid sizes than seen during training for each grid size (Table 2). This suggests TSLM's structured tree representation learns generalizable navigation patterns rather than memorizing specific configurations.

## 6.3 TEST-TIME SCALING: BEYOND PARALLEL SAMPLING

Beyond the quantitative improvements shown in §5.3, TSLM's test-time scaling reveals a qualitative shift in how models can control exploration. Traditional scaling methods treat the model as a fixed sampler: increasing k simply draws more independent samples from the same learned distribution. TSLM scaling is fundamentally different: the model explicitly generates branching structure, enabling adaptive control over which regions of the search space to explore. This transforms scaling from "sample more" to "explore strategically."

The robustness of TSLM's internalized search extends across different base models. Appendix §E.2 demonstrates that TSLM maintains consistent improvements over baselines regardless of the underlying architecture (Llama-3-8B, Llama-3-8B-Instruct, Qwen-2.5-7B), showing that the benefits stem from structured exploration rather than model-specific quirks.

More importantly, TSLM enables flexible search strategies unavailable to parallel sampling methods. By generating an explicit tree structure, we can choose between breadth-first search (prioritizing solution optimality) and depth-first search (prioritizing model confidence), or implement adaptive strategies that expand high-reward branches more aggressively. As shown in Appendix §E.1, DFS achieves 63.1% top-1 accuracy (vs BFS 61.3%) by immediately following high-confidence actions, while BFS provides better coverage for top-k exploration. This controllability is impossible with independent sampling, where each trajectory commits to a full path without coordinating exploration priorities.

Finally, TSLM's scaling extends beyond the training branching factor. TSLM's scaling extends beyond the training branching factor (Appendix §E.3). Figure 6 demonstrates that models trained with k=5 successfully extrapolate to k=10 at inference time, improving convergence from 67.2% to 71.1%. The model generalizes its learned exploration strategy to broader search, unlike parallel sampling which simply draws more independent samples from the same distribution.

## 7 CONCLUSION

We introduce Tree-Structured Language Modeling (TSLM), which enables language models to generate complete search trees within a single generation process using token-based serialization.

TSLM achieves superior performance across structured planning and open-ended reasoning tasks: 100% accuracy on Game of 24 (vs. 17% for baselines), robust extrapolation to larger environments (76.5% vs. 26% for Tree-of-Thought).

Our work challenges the prevailing assumption that reasoning capabilities require reinforcement learning or inference-time search orchestration. By training on complete tree-structured traces rather than single solution paths, supervised learning can internalize systematic exploration strategies. Key advantages over existing approaches include: (1) coherent tree generation vs fragmented parallel sampling, (2) selective context decoupling enabling efficient branch exploration, (3) emergent capabilities like unsolvable problem detection, and (4) a new test-time scaling paradigm that explores within structured trees rather than across independent trajectories.

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

## A  FREQUENTLY ASKED QUESTIONS

**Q1: What do these results mean for reasoning model development more broadly?**  Our results challenge three prevailing assumptions. First, TSLM (67.2%) outperforms ToT (Yao et al., 2023) (62.3%) despite ToT using 100 independent samples, revealing that coherent tree generation beats fragmented parallel sampling. The bottleneck is not inference compute but systematic search space construction. Second, ToT's catastrophic failure on 20×20 Gridworld (42.7%) versus TSLM's extrapolation (91.5%) shows external search does not generalize when complexity scales; the search procedure must be internalized during training. Third, TSLM achieves perfect Game of 24 accuracy via supervised learning while GRPO (Shao et al., 2024) achieves only 15%, challenging the "SFT Memorizes, RL Generalizes" paradigm (Chu et al., 2025). When supervision includes complete exploration traces rather than just answers, supervised learning can acquire systematic reasoning capabilities typically associated with RL methods.

**Q2: How does TSLM differ fundamentally from existing approaches (ToT, standard CoT, GRPO)?** TSLM generates coherent tree structures within a single model forward pass, unlike ToT (independent parallel sampling), standard CoT (single linear path), or GRPO (RL-based trace generation). The key is selective context decoupling: TSLM conditions each branch only on relevant ancestors and siblings, not the entire exploration history. This enables both efficient training (learning from complete trees) and efficient inference (avoiding redundant prefix computation).

**Q3: Can TSLM be applied to more complex real-world tasks?** While our current experiments focus on relatively constrained tasks, the principles of TSLM can extend to more complex domains. For tasks with well-defined structure (like code generation or game playing), direct application is straightforward. For more open-ended tasks, our bootstrapping approach provides a foundation for constructing synthetic tree-structured training data. Future work should explore applications to more diverse and complex reasoning domains.

**Q4: Why compare only with ToT and not advanced methods like RAP, LATS, or Reflexion?** These methods represent online planning (external feedback loops during inference), while TSLM represents offline planning (internalizing search during training). ToT is the representative baseline for external scaffolding without environment interaction. Methods like LATS(Zhou et al., 2023) and Reflexion(Shinn et al., 2023) are orthogonal and could potentially enhance TSLM, but address different problems (online refinement vs native tree generation). Note that our ToT@100 evaluation upper-bounds related sampling methods: mathematically $E[P(\text{RAP@1}) \leq E[P(\text{ToT@}b^d)]$ and $E[P(\text{Self-consistency@}b^d) \leq E[P(\text{ToT@}b^d)]$ since tree expansion has the same expectation as parallel sampling

**Q5: Does TSLM work on truly open-ended tasks?** TSLM relies on the ability to construct training trees, which requires verifiable solutions (even if only for a subset of data). For truly open-ended domains with subjective correctness (e.g., creative writing) or no clear verification, constructing the training signal is challenging. This is a limitation of the current bootstrapping approach, though the core principle of structured exploration may still apply if suitable reward signals can be defined.

# B  RELATED WORK

**External Search-Augmented Language Models.** A major line of work augments language models with external search algorithms during inference. Tree-of-Thought Yao et al. (2023) applies breadth-first and depth-first search externally, sampling multiple reasoning paths with external evaluation. Graph-of-Thought Besta et al. (2024) extends this to general graph structures. Monte Carlo Tree Search (MCTS) approaches include RAP (Reasoning via Planning) Hao et al. (2023), which employs MCTS with world models, and LATS (Language Agent Tree Search) Zhou et al. (2023), which combines MCTS with reflection mechanisms. TS-LLM Feng et al. (2024) and AlphaCode-style approaches Li et al. (2022) integrate MCTS with language generation. These methods achieve strong performance but require multiple model invocations and external orchestration, limiting efficiency and integration with model training.

**Multi-Path Generation and Reasoning.** Various approaches explore multiple reasoning paths without structured search. Self-consistency decoding Wang et al. (2023) generates multiple independent reasoning paths and selects the most consistent answer. Ensemble methods combine predictions from multiple reasoning chains through diverse beam search and nucleus sampling variants. Recent models like o1 OpenAI (2024) and DeepSeek-R1 DeepSeek-AI et al. (2025) generate extended reasoning traces, but remain fundamentally sequential and may include redundant computation Chen et al. (2025). These approaches explore multiple paths but typically generate them independently without capturing structural relationships, unlike TSLM's coherent tree structures.

**Learning-Based Reasoning Enhancement.** Reinforcement learning has emerged as a dominant paradigm for improving reasoning capabilities. DeepSeek Math Shao et al. (2024) applies GRPO for mathematical reasoning, while other work uses RL from human feedback for instruction following. Actor-Critic methods and policy gradient approaches have shown promise in multi-step reasoning and mathematical problem solving. Algorithmic reasoning approaches train models to imitate procedures

like sorting, graph traversal, and dynamic programming Kim et al. (2024); Yang et al. (2022), learning to execute classical algorithms step-by-step. However, these methods either require complex RL training or focus on single algorithmic traces rather than dynamic exploration strategies. TSLM demonstrates that carefully structured supervised learning can achieve comparable performance without RL's complexity.

## C  TSLM Serialization Example

This section provides a detailed worked example showing how TSLM serializes tree structures for the Game of 24 task.

**Problem Setup.**  Consider a Game of 24 problem with numbers [4, 5, 6, 10]. The goal is to find arithmetic operations that result in 24.

**Complete Tree Serialization.**  The tree structure is serialized as:

```
Input:
4 + 5 = 9 [SEP]
6 + 9 = 15 [SEP]

Output:
[BOS]
9 + 15 = 24 [GOAL]
9 – 15 = –6 [FAIL] [EOS]
```

This format captures:

- Multiple candidate actions at each step
- Viability markers (`[SEP]` for expandable, `[FAIL]` for dead ends)
- Step boundaries (`[BOS], [EOS]`)
- Both successful and unsuccessful exploration paths

## D  Limitations and Future Work

**Computational Overhead.**  Training on complete trees requires processing all nodes rather than single paths, increasing computational cost by a factor of the average tree size. Future work should explore efficient training strategies like tree-aware attention caching and sparse gradient updates for selective subtree optimization.

**Supervision Quality Dependence.**  Synthetic tree generation quality depends heavily on the supervision model's capabilities. For instance, Llama-3-8B produces poor-quality trees for GSM8K, while Llama-3-8B-Instruct works well. Models must match the target reasoning format (some generate Python code instead of step-by-step reasoning, others include extraneous dialogue). Better automated methods for supervision model selection and tree quality validation are needed.

**Open-Ended Domains.**  TSLM requires verifiable solution paths to construct training trees. In truly open-ended domains with subjective correctness (creative writing, open dialogue), defining branch viability becomes challenging. Adapting TSLM to such domains may require preference learning or soft verification signals rather than binary correctness.

## E  Detailed Inference-Time Scaling Analysis

### E.1  BFS vs DFS: Exploring Search Strategies

Within TSLM's inference framework, the choice between Breadth-First Search (BFS) and Depth-First Search (DFS) reveals fundamental differences in exploration priorities. We analyze these search strategies when TSLM's branching is guided by preference-based ordering.

**Theoretical Analysis.** Consider two distinct solution paths $A = a_1, ..., a_n$ and $B = b_1, ..., b_m$ in a preference-ordered search tree generated by TSLM.

BFS employs a dual prioritization strategy: it first considers path length (traversing $A$ before $B$ if $n < m$), then uses preference signals to break ties. When paths have equal length ($n = m$), BFS selects based on the first differing action's reward. If $R(a_k) > R(b_k)$ at the earliest divergence point $k$, BFS favors path $A$. This means **BFS prioritizes solution optimality first, using learned preferences to resolve ties**.

In contrast, DFS operates purely on learned preference ordering, disregarding path length considerations. For the same paths $A$ and $B$, DFS immediately follows the higher-reward action at any divergence point $k$, regardless of whether this leads to longer solution paths. This approach means **DFS prioritizes learned preferences consistently, potentially sacrificing optimality for high-confidence actions**.

**Empirical Results.** Figure 4 shows that DFS achieves better top-1 accuracy by finding preferred solutions first, but slightly underperforms BFS for top-3 to top-6 candidates since it does not prioritize optimality. However, both methods converge to the same accuracy as they traverse the complete tree. This validates our theoretical analysis that DFS provides faster convergence to high-confidence solutions while BFS offers better overall exploration coverage.

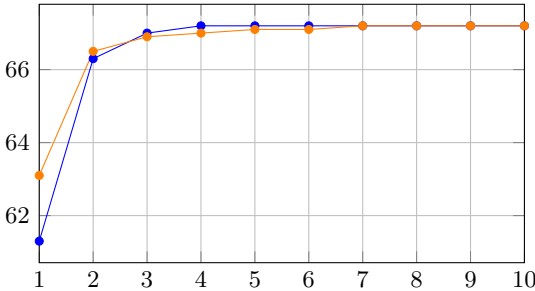

Figure 4: Comparison of mean accuracy (%) over number of candidates for different search strategies (BFS: •, DFS •)

### E.2 BASE MODEL PERFORMANCE IMPACT

We investigate whether the base model's performance influences TSLM's effectiveness by comparing performance across different models with different GSM8K performance (Llama-3-8B, Llama-3-8B-Instruct, and Qwen-2.5-7B). TSLM consistently outperforms baseline methods across both model variants, with improvements remaining robust across architectures. This indicates that TSLM's structured exploration provides consistent benefits independent of the base model.

### E.3 BRANCHING FACTOR ANALYSIS

The branching factor $k$ in search tree generation (Algorithm 1) controls the maximum number of candidates expanded at each node. We initially set $k = 5$ to match the supervision breadth, but a natural question arises: Can TSLM effectively scale to larger branching factors?

In principle, if we traverse the complete search tree, a larger $k$ should yield better accuracy by exploring more candidates. However, when limited to examining just the first few terminated candidates, increasing the branching factor could potentially add noise that degrades performance.

We investigate this empirically by comparing TSLM performance between $k = 5$ and $k = 10$ branching factors. As shown in Figure 6, extrapolating to the larger branching factor yields consistent improvements, with convergence rate increasing from 67.2% to 71.1%. This enhanced performance indicates TSLM can effectively leverage broader exploration without being overwhelmed by the expanded search space.

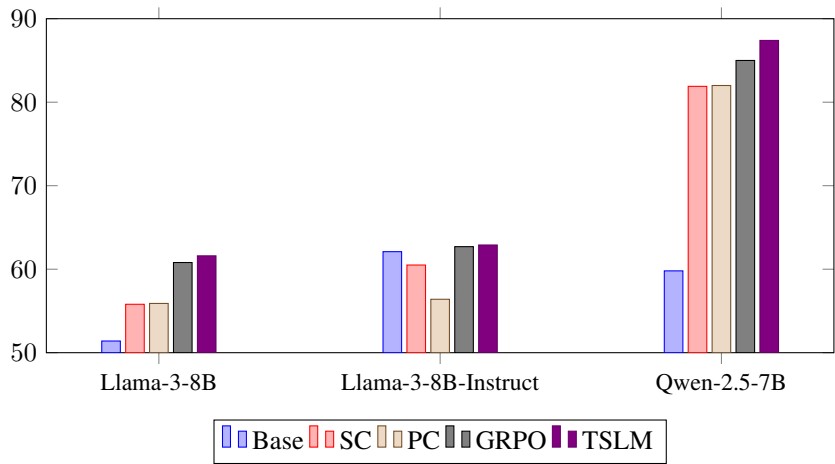

Figure 5: GSM8K accuracy comparison across different base models and methods.

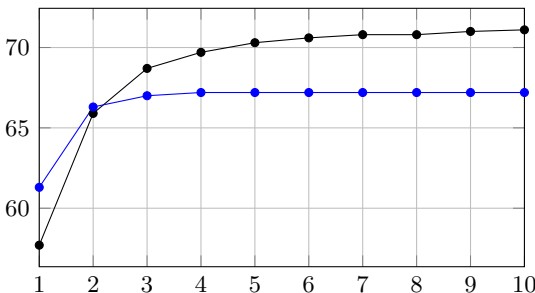

Figure 6: Comparison of TSLM performance for the base branching factors $k = 5$ (●) vs. extrapolated $k = 10$ (●). Extrapolating the branching factor improves accuracy in a scalable manner.

## F    TASK DETAILS AND EXAMPLES

| Task | # Depth(Train) | # Depth(Test) | # Train Instance | # Test Instance | Evaluation Metric |
|------|----------------|---------------|------------------|-----------------|-------------------|
| Game of 24 | 3 | 3 | 1.2K | 100 | Equation Validation |
| Gridworld | $10 \times 10$ | $10 \times 10$ | 10K | 1.5K | Exact Matching |
| Gridworld (o.o.d) | $10 \times 10$ | $20 \times 20$ | 10K | 1.5K | Exact Matching |
| ProntoQA | 1-5 | 1-5 | 4.5K | 450 | Exact Matching |
| GSM8K | 2-9 | 2-11 | 7.5K | 1.3K | Answer Matching |

Table 3: General task settings

### F.1    GENERAL TASK SETUP DETAILS

Table 3 illustrates the general task settings for structured and open-ended tasks. Depth refers to the number of reasoning steps.

### F.2    TASK SCENARIO 1: STRUCTURED PLANNING

We first evaluate TSLM and baselines on two different structured planning tasks (Left in Figure 7). These tasks involve predefined search trees, enabling us to evaluate how well models reproduce algorithmic patterns:

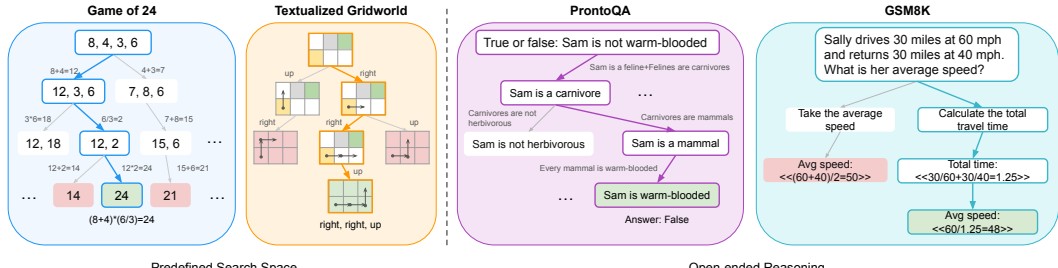

Figure 7: (Left) Structured planning tasks with predefined search spaces (e.g., Game of 24, Textualized Gridworld) where success is measured by the model's ability to reproduce algorithmic search patterns. (Right) General open-ended reasoning tasks (e.g., ProntoQA, GSM8K) requiring adaptive exploration of undefined search spaces.

**Game of 24** Given four numbers, the task is to obtain 24 with basic arithmetic operations $(+, -, \times, \div)$. For example, with inputs 8, 4, 3, and 6, a valid solution is $(8 + 4) \times (6 \div 3) = 24$ Yao et al. (2023). Each action selects two operands and an operator, while states represent partial expressions. Since there may be multiple paths that reaches the answer (e.g. $(8 + 4) \times (6 \div 3)$ and $(6 \div 3) \times (4 + 8)$), we check if the final expression equals 24. We train each method on 1.2K instances and test on 100 instances.

**Textualized Gridworld** A text-based navigation task where an agent navigates a grid using cardinal directions (up, down, left, right) while avoiding obstacles (Kim et al., 2024). Given a start position (typically bottom left) and goal position (typically top right), the agent must find a valid path with shortest length. For instance, in a 3x2 grid with a pit at (1,1), the optimal solution is (right, right, up). Actions are single moves (up/down/left/right) and states are grid coordinates. We ensure all the environments to have unique shortest paths. We use 10K training instances with maximum grid size 10x10 and 1.5K test instances up to the same maximum size, evaluating exact path matching.

### F.3 Task Scenario 2: Open-ended Reasoning Tasks

We next evaluate TSLM and baselines on two open-ended reasoning tasks (Right on Figure 7) requiring exploration of undefined solution spaces. We use the synthetic tree generation method from §4.2 for training.

**ProntoQA** ProntoQA Saparov & He (2023) is a logical reasoning dataset where models verify statement truth given premises. For example, given a premise "Every jompus is not small. Each impus is small. Each jompus is a dumpus. Alex is a jompus." and a query "True or false: Alex is not small?", the solution path to verify the query should be "Alex is a jompus. Every jompus is not small. Alex is not small. So the answer is True."

Actions are logical deduction steps and states are accumulated facts. With unique valid deduction paths, we verify whether the generated path is correct. We use 4.5K training and 450 test instances. For TSLM/PC, we use tree generation of beam search (k=5, temp=0.3) with Llama-3-8B as a base model for supervision.

**GSM8K** GSM8K Cobbe et al. (2021) is a mathematical word problem dataset requiring multi-step reasoning. For example, for a given problem "James runs 3 sprints 3 times a week. Each sprint is 60 meters. How many total meters per week?", one valid solution would be "3 sprints × 3 times = 9 sprints per week. 9 sprints × 60 meters = 540 meters. The answer is 540."

Actions are solution steps and states are partial solutions. Since multiple valid solution paths can reach the correct answer, we just validate whether the answer is correct. We use 7.5K training instances and 1.3K test instances. For TSLM/PC, we use tree generation of beam search (k=5, temp=0.1) with Llama-3-8B-Instruct as a base model for supervision.

### F.4 Task Examples

**Gridworld SC Instance:**

**USER:** You are given a rectangular gridworld, where you can move up, down, left, or right as long as each of your x, y coordinate is within 0 to the x, y size of the grid. If you move up, your y coordinate increases by 1. If you move down, your y coordinate decreases by 1. If you move left, your x coordinate decreases by 1. If you move right, your x coordinate increases by 1.

You will interact with the girdworld environment to reach the goal state, while avoiding the pit and the wall. You cannot move through the wall or move outside the grid. If you fall into the pit, you lose. If you reach the goal, you win. For each of your turn, you will be given the possible moves.

You should respond your move with either one of 'up', 'down', 'left', or 'right'.

**ASSISTANT:** OK

**USER:** Grid is from (7, 7) to (18, 13). Goal: (18, 13) Current: (7, 7) The pit is at (16, 9), (17, 7), (14, 8), (11, 9), and (16, 11). The wall is at (11, 7), (15, 10), (18, 12), (9, 9), (8, 7), (17, 10), (9, 12), (12, 7), (10, 13), (15, 12), (17, 9), (9, 11), (17, 12), (8, 9), (13, 11), (10, 9), (12, 9), (11, 11), (12, 12), (16, 10), (15, 8), (9, 7), (8, 11), (7, 13), (13, 13), and (13, 10).

**ASSISTANT:**

Step 1
up
(7, 8)[SEP]
Step 2
up
(7, 9)[SEP]
Step 3
up
(7, 10)[SEP]
Step 4
up
(7, 11)[SEP]
...

**Gridworld PC Instance:**

**USER:** You are given a rectangular gridworld, where you can move up, down, left, or right as long as each of your x, y coordinate is within 0 to the x, y size of the grid. If you move up, your y coordinate increases by 1. If you move down, your y coordinate decreases by 1. If you move left, your x coordinate decreases by 1. If you move right, your x coordinate increases by 1.

You will interact with the girdworld environment to reach the goal state, while avoiding the pit and the wall. You cannot move through the wall or move outside the grid. If you fall into the pit, you lose. If you reach the goal, you win. For each of your turn, you will be given the possible moves.

You should respond your move with either one of 'up', 'down', 'left', or 'right'.

**ASSISTANT:** OK

**USER:** Grid is from (7, 7) to (18, 13). Goal: (18, 13) Current: (7, 7) The pit is at (16, 9), (17, 7), (14, 8), (11, 9), and (16, 11). The wall is at (11, 7), (15, 10), (18, 12), (9, 9), (8, 7), (17, 10), (9, 12), (12, 7), (10, 13), (15, 12), (17, 9), (9, 11), (17, 12), (8, 9), (13, 11), (10, 9), (12, 9), (11, 11), (12, 12), (16, 10), (15, 8), (9, 7), (8, 11), (7, 13), (13, 13), and (13, 10).

**ASSISTANT:**

Step 1
up
(7, 8)[SEP]
down
(7, 6)[FAIL]
left
(6, 7)[FAIL]
right
(8, 7)[FAIL]
Step 2

up
(7, 9)[SEP]
down
(7, 7)[FAIL]
left
(6, 8)[FAIL]
right
(8, 8)[SEP]
...

**Gridworld TSLM Instance:**
**USER:** You are given a rectangular gridworld, where you can move up, down, left, or right as long as each of your x, y coordinate is within 0 to the x, y size of the grid. If you move up, your y coordinate increases by 1. If you move down, your y coordinate decreases by 1. If you move left, your x coordinate decreases by 1. If you move right, your x coordinate increases by 1.
You will interact with the girdworld environment to reach the goal state, while avoiding the pit and the wall. You cannot move through the wall or move outside the grid. If you fall into the pit, you lose. If you reach the goal, you win. For each of your turn, you will be given the possible moves.
You should respond your move with either one of 'up', 'down', 'left', or 'right'.
**ASSISTANT:** OK
**USER:** Grid is from (7, 7) to (18, 13). Goal: (18, 13) Current: (7, 7) The pit is at (16, 9), (17, 7), (14, 8), (11, 9), and (16, 11). The wall is at (11, 7), (15, 10), (18, 12), (9, 9), (8, 7), (17, 10), (9, 12), (12, 7), (10, 13), (15, 12), (17, 9), (9, 11), (17, 12), (8, 9), (13, 11), (10, 9), (12, 9), (11, 11), (12, 12), (16, 10), (15, 8), (9, 7), (8, 11), (7, 13), (13, 13), and (13, 10).
**ASSISTANT:**
Step 1
up
(7, 8)[SEP]
Step 2
up
(7, 9)[SEP]
Step 3
up
(7, 10)[SEP]
Step 4
up
(7, 11)[SEP]
down
(7, 9)[FAIL]
left
(6, 10)[FAIL]
right
(8, 10)[SEP]

**Game24 SC Instance:**
**USER:** Input: 4, 5, 6, 10
**ASSISTANT:** Step 1
4 + 5 = 9[SEP]
Step 2
6 + 10 = 16[SEP]

**Game24 PC Instance:**
**USER:** Input: 4, 5, 6, 10
**ASSISTANT:** Step 1
4 + 5 = 9[SEP]
4 - 5 = -1[SEP]
4 * 5 = 20[SEP]
4 / 5 = 4/5[SEP]

4 + 6 = 10[SEP]
4 - 6 = -2[SEP]
4 * 6 = 24[SEP]
4 / 6 = 2/3[SEP]
4 + 10 = 14[SEP]
4 - 10 = -6[SEP]
4 * 10 = 40[SEP]
4 / 10 = 2/5[SEP]
5 + 4 = 9[SEP]
5 - 4 = 1[SEP]
5 * 4 = 20[SEP]
5 / 4 = 5/4[SEP]
5 + 6 = 11[SEP]
5 - 6 = -1[SEP]
5 * 6 = 30[SEP]
5 / 6 = 5/6[SEP]
5 + 10 = 15[SEP]
5 - 10 = -5[SEP]
5 * 10 = 50[SEP]
5 / 10 = 1/2[SEP]
6 + 4 = 10[SEP]
6 - 4 = 2[SEP]
6 * 4 = 24[SEP]
6 / 4 = 3/2[SEP]
6 + 5 = 11[SEP]
6 - 5 = 1[SEP]
6 * 5 = 30[SEP]
6 / 5 = 6/5[SEP]
6 + 10 = 16[SEP]
6 - 10 = -4[SEP]
6 * 10 = 60[SEP]
6 / 10 = 3/5[SEP]
10 + 4 = 14[SEP]
10 - 4 = 6[SEP]
10 * 4 = 40[SEP]
10 / 4 = 5/2[SEP]
10 + 5 = 15[SEP]
10 - 5 = 5[SEP]
10 * 5 = 50[SEP]
10 / 5 = 2[SEP]
10 + 6 = 16[SEP]
10 - 6 = 4[SEP]
10 * 6 = 60[SEP]
10 / 6 = 5/3[SEP][FAIL]
Step 2
6 + 10 = 16[SEP]
6 - 10 = -4[SEP]
6 * 10 = 60[SEP]
6 / 10 = 3/5[SEP]
6 + 9 = 15[SEP]
6 - 9 = -3[SEP]
6 * 9 = 54[SEP]
6 / 9 = 2/3[SEP]
10 + 6 = 16[SEP]
10 - 6 = 4[SEP]
10 * 6 = 60[SEP]
10 / 6 = 5/3[SEP]
10 + 9 = 19[SEP]
10 - 9 = 1[SEP]

```
10 * 9 = 90[SEP]
10 / 9 = 10/9[SEP]
9 + 6 = 15[SEP]
9 - 6 = 3[SEP]
9 * 6 = 54[SEP]
9 / 6 = 3/2[SEP]
9 + 10 = 19[SEP]
9 - 10 = -1[SEP]
9 * 10 = 90[SEP]
9 / 10 = 9/10[SEP][FAIL]
6 + 10 = 16[SEP]
6 - 10 = -4[SEP]
6 * 10 = 60[SEP]
6 / 10 = 3/5[SEP]
6 + -1 = 5[SEP]
6 - -1 = 7[SEP]
6 * -1 = -6[SEP]
6 / -1 = -6[SEP]
10 + 6 = 16[SEP]
10 - 6 = 4[SEP]
10 * 6 = 60[SEP]
10 / 6 = 5/3[SEP]
10 + -1 = 9[SEP]
10 - -1 = 11[SEP]
10 * -1 = -10[SEP]
10 / -1 = -10[SEP]
-1 + 6 = 5[SEP]
-1 - 6 = -7[SEP]
-1 * 6 = -6[SEP]
-1 / 6 = -1/6[SEP]
-1 + 10 = 9[SEP]
-1 - 10 = -11[SEP]
-1 * 10 = -10[SEP]
-1 / 10 = -1/10[SEP][FAIL]
6 + 10 = 16[SEP]
6 - 10 = -4[SEP]
6 * 10 = 60[SEP]
6 / 10 = 3/5[SEP]
6 + 20 = 26[SEP]
6 - 20 = -14[SEP]
6 * 20 = 120[SEP]
6 / 20 = 3/10[SEP]
10 + 6 = 16[SEP]
10 - 6 = 4[SEP]
10 * 6 = 60[SEP]
10 / 6 = 5/3[SEP]
10 + 20 = 30[SEP]
10 - 20 = -10[SEP]
10 * 20 = 200[SEP]
10 / 20 = 1/2[SEP]
20 + 6 = 26[SEP]
20 - 6 = 14[SEP]
20 * 6 = 120[SEP]
20 / 6 = 10/3[SEP]
20 + 10 = 30[SEP]
20 - 10 = 10[SEP]
20 * 10 = 200[SEP]
20 / 10 = 2[SEP][FAIL]
```

**Game24 TSLM Instance:**
**USER:** Input: 4, 5, 6, 10
**ASSISTANT:** Step 1
4 + 5 = 9[SEP]
Step 2
6 + 10 = 16[SEP]
6 - 10 = -4[SEP]
6 * 10 = 60[SEP]
6 / 10 = 3/5[SEP]
6 + 9 = 15[SEP]
6 - 9 = -3[SEP]
6 * 9 = 54[SEP]
6 / 9 = 2/3[SEP]
10 + 6 = 16[SEP]
10 - 6 = 4[SEP]
10 * 6 = 60[SEP]
10 / 6 = 5/3[SEP]
10 + 9 = 19[SEP]
10 - 9 = 1[SEP]
10 * 9 = 90[SEP]
10 / 9 = 10/9[SEP]
9 + 6 = 15[SEP]
9 - 6 = 3[SEP]
9 * 6 = 54[SEP]
9 / 6 = 3/2[SEP]
9 + 10 = 19[SEP]
9 - 10 = -1[SEP]
9 * 10 = 90[SEP]
9 / 10 = 9/10[SEP][FAIL]

**ProntoQA SC Instance:**
**USER:** Arthropods are small. Insects are not eight-legged. Lepidopterans are insects. Invertebrates are animals. Every spider is eight-legged. Insects are arthropods. Every butterfly is a lepidopteran. Animals are multicellular. Arthropods are invertebrates. Sam is a butterfly.
True or false: Sam is not eight-legged.
**ASSISTANT:**
Step 1
Sam is a butterfly.[SEP]
Step 2
Every butterfly is a lepidopteran.[SEP]
Step 3
Sam is a lepidopteran.[SEP]
Step 4
Lepidopterans are insects.[SEP]
Step 5
Sam is an insect.[SEP]
Step 6
Insects are not eight-legged.[SEP]
Step 7
Sam is not eight-legged.[SEP]
Step 8
The answer is true.[GOAL]

**ProntoQA PC Instance:** c Arthropods are small. Insects are not eight-legged. Lepidopterans are insects. Invertebrates are animals. Every spider is eight-legged. Insects are arthropods. Every butterfly is a lepidopteran. Animals are multicellular. Arthropods are invertebrates. Sam is a butterfly.
True or false: Sam is not eight-legged.
**ASSISTANT:**

Step 1
Sam is a butterfly.[SEP]
Step 2
Every butterfly is a lepidopteran.[SEP]
Step 3
Sam is a lepidopteran.[SEP]
Step 4
Lepidopterans are insects.[SEP]
Step 5
Sam is an insect.[SEP]
Step 6
Insects are not eight-legged.[SEP]
Insects are arthropods.[SEP]
Step 7
Sam is not eight-legged.[SEP]
Sam is an arthropod.[GOAL]

**ProntoQA TSLM Instance:**
**USER:** Arthropods are small. Insects are not eight-legged. Lepidopterans are insects. Invertebrates are animals. Every spider is eight-legged. Insects are arthropods. Every butterfly is a lepidopteran. Animals are multicellular. Arthropods are invertebrates. Sam is a butterfly.
True or false: Sam is not eight-legged.
**ASSISTANT:**
Step 1
Sam is a butterfly.[SEP]
Step 2
Every butterfly is a lepidopteran.[SEP]
Step 3
Sam is a lepidopteran.[SEP]
Step 4
Lepidopterans are insects.[SEP]
Step 5
Sam is an insect.[SEP]
Step 6
Insects are not eight-legged.[SEP]
Step 7
Sam is not eight-legged.[SEP]
Sam is an arthropod.[GOAL]

**GSM8K SC Instance:**
**USER:** Kylar went to the store to buy glasses for his new apartment. One glass costs $5, but every second glass costs only 60% of the price. Kylar wants to buy 16 glasses. How much does he need to pay for them?
**ASSISTANT:**
Step 1
Every second glass costs $5 * 0.6 = \$ << 5 * 0.6 = 3 >> 3$.[SEP]
Step 2
So for 16 glasses, Kylar needs to pay $3 * 8 = \$ << 16/2 = 8 >>$ for every second glass.[GOAL]

**GSM8K PC Instance:**
**USER:** A robe takes 2 bolts of blue fiber and half that much white fiber. How many bolts in total does it take?
**ASSISTANT:**
Step 1
It takes $2 * 0.5 = << 2 * 0.5 = 1 >> 1$ bolt of white fiber.[SEP]
The robe takes 2 bolts of blue fiber.[SEP]
Step 2
So it takes $2 + 1 = << 2 + 1 = 3 >> 3$ bolts in total.[SEP]

It also takes half as much white fiber, which means it takes 1 bolt of white fiber (since half of 2 is 1).[SEP]
Step 3
The answer is 3.[GOAL]

---

**GSM8K TSLM Instance:**
**USER:** Kylar went to the store to buy glasses for his new apartment. One glass costs $5, but every second glass costs only 60% of the price. Kylar wants to buy 16 glasses. How much does he need to pay for them?
**ASSISTANT:**
Step 1
Every second glass costs $5 * 0.6 = \$ << 5 * 0.6 = 3 >> 3$.[SEP]
Step 2
So for 16 glasses, Kylar needs to pay $3 * 8 = \$ << 16/2 = 8 >>$ for every second glass.[SEP]
So for 8 glasses, it would cost $8 \times 3 = 24$ dollars. For the remaining 8 glasses, it would cost $8 \times 5 = 40$ dollars. So in total, it would cost $24 + 40 = 64$ dollars.[GOAL]

## G    USE OF LARGE LANGUAGE MODELS

We acknowledge the use of large language models to assist in the preparation of this manuscript. Specifically:

**Writing assistance.**    Large language models were used to aid in polishing and refining the writing throughout the paper, including improving clarity, grammar, and expression of technical concepts.

**Related work discovery.**    Large language models were employed for retrieval and discovery tasks, particularly in identifying and organizing relevant related work and ensuring comprehensive coverage of the literature.

All technical contributions, experimental design, implementation, analysis, and conclusions presented in this work are the original work of the authors. The use of LLMs was limited to editorial assistance and literature search support, and did not influence the core scientific contributions or findings reported in this paper.

## H    ACKNOWLEDGEMENT

This work was partly supported by Institute of Information  communications Technology Planning  Evaluation (IITP) grant funded by the Korea government (MSIT) (No.RS-2019-II190075 Artificial Intelligence Graduate School Program (KAIST), 10%; No.RS-2021-II212068, Artificial Intelligence Innovation Hub, 10%; RS-2024-00398115, Research on the reliability and coherence of outcomes produced by Generative AI, 20%; No.2022-0-00113, Developing a Sustainable Collaborative Multi-modal Lifelong Learning Framework, 20%; No.RS-2022-II220264, Comprehensive Video Understanding and Generation with Knowledge-based Deep Logic Neural Network, 20%; RS-2024-00397966, Development of a Cybersecurity Specialized RAG-based sLLM Model for Suppressing Gen-AI Malfunctions and Construction of a Publicly Demonstration Platform) and the InnoCORE program of the Ministry of Science and ICT(N10250156).

