# OpenReview forum: "TSLM: Tree-Structured Language Modeling for Divergent Thinking"
_ICLR.cc/2026/Conference — ICLR 2026 Poster_

### Official Review · Reviewer_Aewk · 2025-10-28

**Soundness:** 3
**Presentation:** 3
**Contribution:** 3
**Rating:** 4
**Confidence:** 3

**Summary:**

The paper proposes Tree-Structured Language Modeling (TSLM), a simple but distinctive way to let a standard transformer natively generate and traverse a search tree within a single sequence. The core idea is to serialize a tree into a linear token stream using special markers, so that teacher traces include both successful and failed branches. Training is standard LM loss on these serialized traces; inference reconstructs the tree and explores it until a solution or exhaustion. The paper reports results across Game of 24, Textualized Gridworld, ProntoQA, and GSM8K. On structured tasks, TSLM is very strong, shows rapid adaptation with few samples on ProntoQA, and claims parameter efficiency. For open-ended math (GSM8K), TSLM is competitive. Overall, TSLM is positioned as a purely supervised alternative to RL-style “reasoning” methods that require external search or test-time orchestration.

**Strengths:**

S1: The token-based tree serialization for end-to-end supervised training of internal exploration is a neat, low-friction idea. It avoids extra modules (verifiers or value functions) and RL, and while LM-based search is well-studied, representing divergent branches natively within a single generation is an original framing with distinct trade-offs.

S2: Strong performance on Game of 24 and Gridworld demonstrates the method’s core promise. The BFS vs. DFS analysis in Appendix G provides insight into how traversal strategies interact with learned preferences.

S3. No architectural changes are required; the approach works with standard SFT and any base LM (e.g., Llama-3, Qwen). Training and inference procedures are clear, with many worked examples provided, supporting reproducibility.

**Weaknesses:**

W1: ToT is configured minimally, and comparisons omit Self-Consistency and verifier-based selection, which are strong baselines on GSM8K, as well as natural tree-search comparators like TS-LLM, LATS, and RAP. Without these, it is difficult to judge when TSLM’s internalized tree search provides meaningful benefits.


W2: TSLM trains over all tree nodes O(N⋅L)), and inference produces long serialized trees. Claims of efficiency are not quantitatively compared against ToT’s extra model calls or SC/self-consistency multi-sample decoding.

W3: The tasks are well-suited to structured search-tree reasoning but do not represent open-ended domains such as large-scale software engineering, real-world system design, multi-agent planning, or long-horizon code generation. Claims that strong performance on structured tasks will generalize to these domains remain untested and speculative.

**Questions:**

Q1. What were the exact ToT settings (beam width, evaluation heuristic, depth/budget) per task? Did you evaluate Self-Consistency (sample-and-vote) or GSM8K verifier selection? If so, how do they compare under equal token budgets?

Q2: The reported scaling performance (“68% for 0.5B vs 19–26% for 7B baselines”) requires clarification: What is the exact metric and task (Gridworld 20×20 only?), and are these single-seed results? Please provide confidence intervals.

Q3: The paper provides a qualitative example. Across a large set, how often does TSLM correctly abstain on unsolvable Game of 24 instances?

Q4: How well does TSLM handle branches in open-ended problem domains, where the branching structure is unclear or multiple valid solution paths exist without a single “correct” branch?

---

> ### Author Response · Authors · 2025-11-20
>
> Thank you for the detailed and thoughtful review. We appreciate your recognition of TSLM. Let us address your concerns directly, but before that, we strongly recommend reading the general comment first, since it contains important context regarding your comment.
>
> # W1, W2, & Q1: Baseline Comparisons and ToT Configuration
>
> The ToT evaluation of our experiment (ToT@100) is different from (actually more loosen than) the original ToT paper (ToT@1, Yao et al.). The general comment explains this in detail, but the key insight is:
>
> ToT in our experiments is pass@100, while TSLM and all other baselines are pass@1. ToT examines up to 100 terminal nodes via BFS exploration (operationally comparable to pass@100, though with higher variance due to correlated paths in the tree structure). This makes direct comparison misleading. For fair comparison at matched fixed budget, see Rebuttal Table 2 in the general comment. TSLM@5 achieves 67.2% on GSM8K, while ToT needs to scale to ToT@100 just to reach 62.3%. This demonstrates TSLM converges faster with fewer attempts. _We view this as an efficiency coming from the model’s ability to causally generate the search tree rather than redundant sampling from scaffolding._
>
>
> **For ToT or RAP:** **The ToT@100 evaluation already creates an upper bound for related sampling approaches like ToT@1 (Yao et al.), RAP@1 (Hao et al.), or Self-consistency@100 (Wang et al.).** Mathematically, we can establish both  E[P(ToT@1)] ≤ E[P(ToT@b^d)] and E[P(RAP@1)] ≤ E[P(ToT@b^d)] (ToT@b^d passes the instance if one of the whole search tree with beam width b and depth d passes). Also, E[P(Self-consistency@b^d)] ≤ E[P(Pass@b^d)] =E[P(ToT@b^d)], since tree expansion has the same expectation with parallel sampling. We empirically found that E[P(ToT@100)] already converges in our setting, so we used **ToT@100 as a unified upper bound for the whole tree search scaffolding methods**.
>
>
> **For self-refinement methods like TS-LLM or LATS:**
> We want to clarify an important conceptual distinction between these and our methods:
>
> * **Training LMs for offline planning (our focus): SC, PC, GRPO, TSLM** - these aim to train the model to generate search trees for problems presented at inference time, like what reasoning models like o1 or Deepseek-R1 are doing.
>
>
> * **Scaffolding for offline planning (which we have for baseline): ToT, RAP** - these apply external search algorithms on top of a fixed base model (scaffolding), which we show not to be as good as the first line of methods (see #3 of the general comment)
>
>
> * **Self-refinement for online planning: Reflexion, LATS, xoT** - these methods integrate information from interaction/feedback to iteratively revise reasoning
>
>
> **Our work targets offline planning capability;** Teaching the model to construct good search trees when first seeing a problem. The self-refinement methods you mentioned address online planning, which revises solutions based on external feedback or execution results. While these appear similar (both involve exploration), they serve different purposes. For instance, in code generation, offline planning means generating well-structured code initially, while online planning means debugging code based on test failures.
>
>
> Online planning methods could potentially enhance offline capability (by having the model self-revise without external feedback), but it’s more natural to consider this **orthogonal** to our current contribution.”
>
>
> # Q2: Scaling Performance Clarification
>
>
> We followed the same setup which “How language models extrapolate outside the training data: A case study in textualized gridworld” (Kim et al.) used. We train the model with Gridworld solving demonstrations _up to_ 10x10 grid (10K instances), and testing on environments with size _up to_ 20x20 (for Table 2, 1.5K instances). The metric showed in Table 1 is the aggregated success rate (exact path matching rate across all test instances; If you want grid-wise performance then it looks similar to Table 2). We did the single seed result, though it’s safe from cheating since we ensure that the train/test is splitted.

---

> > ### Author Response · Authors · 2025-11-20
> > **Rebuttal cont.**
> >
> > # Q3: Unsolvable Problem Detection
> >
> > We conducted additional experiments on this capability. We manually collected 100 Game of 24 instances with no valid solution and tested each method's ability to correctly abstain from answering:
> >
> > | Game of 24 | SC | PC | GRPO | TSLM (ours) |
> > |---------:|:--------:|----------:|----------:|----------:|
> > |  Verifying unsolvable   | 0  | 3     | 0 | **97**|
> >
> >
> > **TSLM is the only method that reliably terminates without generating an answer.** The other methods cannot refuse to answer because their training and reward design focused exclusively on solving problems, not on detecting unsolvability. The 3 instances where PC correctly refused were unintentional since the model generated excessively long traces that failed to terminate properly due to skewed search tree generation.
> >
> > We position this capability (Section 6.1) as addressing a fundamental limitation of sequential language models. Hallucination remains a major problem in language modeling, with theoretical work suggesting it's inevitable in the next-token prediction paradigm (Xu et al., "Hallucination is Inevitable: An Innate Limitation of Large Language Models"). Frontier labs attempt to address this through post-training to modify refusal behavior (Kalai et al., "Why Language Models Hallucinate"), but the mechanisms remain unclear and heavily rely on RL approaches.
> >
> > We view that TSLM offers a structural advantage here over sequential language modeling paradigm. Because it generates a complete search tree, it can evaluate each terminal node independently to verify solution correctness. When all paths fail verification, the model can confidently determine the problem is unsolvable. Sequential models lack this capability because they commit to a single path and have no mechanism to systematically verify exhaustive exploration iteratively.
> >
> >
> >
> >
> > # W3 & Q4: Open-Ended Domain Generalization
> >
> >
> >
> > We appreciate this concern, but want to reframe what constitutes a meaningful academic contribution. The problem setting you describe (subjective correctness, unclear branching structure, no verifiable rewards) is primarily relevant to frontier labs with massive models and computational resources, and the people who is directly using that model. For academic research, the more pressing question is: can we demonstrate a paradigm shift in how language models learn to reason? Our contribution targets exactly this by showing that **structured supervision fundamentally changes what models can learn, not just incrementally improving performance on existing benchmarks.**
> >
> >
> >
> >
> > Our data requirements match standard approaches: Algorithm 1 needs exactly the same data source as supervised fine-tuning - one correct solution per training example. The bootstrapping procedure guarantees at least one valid solution in every training tree (line 281), then fills the rest with model-generated exploration. In short, if you can train with SFT, you can train with TSLM.
> >
> >
> > Moreover, our requirements are actually less stringent than RL baselines. GRPO needs thousands of environment rollouts per problem to learn from sparse rewards, which is why it collapses to 6% on scaling tasks  (Yes, cold start is brutal when you need massive exploration just to get any signal). We just need one verified solution plus the model's own samples for diversity. As long as we have an objective verifiable reward (the same setting needed for GRPO), TSLM achieves better sample efficiency and higher performance than GRPO.
> >
> >
> > **TSLM combines strengths of both SFT and RL:** When the base model is already good, the search tree naturally converges toward correct paths, making tree structure easier to learn. This is the scenario where people typically use RL over SFT, and TSLM achieves comparable results (see GSM8K on Qwen 2.5: TSLM 87.4% vs GRPO 85.0%). When the base model struggles (like Game of 24 or Gridworld), the tree has many dead ends, but we still guarantee one correct solution so the model learns task-solving. This is where people use SFT over RL, and TSLM still works effectively (and even better on our tasks).
> >
> >
> > Other than the above constrain, our results can extend naturally to open-ended tasks (other than ProntoQA, GSM8K). The tasks branch we chose is at the sentence level, which is a general formulation. The method is also model-agnostic (see general comment #4), showing Algorithm 1 works well across open-source language models.
> >
> > We want to once more emphasize that we are not claiming to solve all reasoning tasks or build the best agent on every possible benchmark.

---

> > > ### Comment · Reviewer_Aewk · 2025-11-24
> > >
> > > Thanks for the authors’ response. I appreciate the effort to address my questions. Regarding the reply to W3 and Q4, I fully understand the resource constraints involved in evaluating additional open scenarios, and I am not asking the authors to test on all of them. My main concern is that the current evaluated tasks, Game of 24 and Gridworld, while useful for demonstration, they are highly idealized, close-form puzzles, and may not fully reflect the complexity, ambiguity, or open-ended reasoning encountered in practical applications. Similarly, GSM8K is relatively simple, and ProntoQA is synthetic in nature, which limits the extent to which the evaluation demonstrates real-world usefulness.
> > >
> > > Even within academic research, claiming a paradigm shift in how language models learn reasoning is a significant and ambitious claim. Relying solely on tasks like Game of 24 or other simple benchmarks makes it difficult to assess whether the method truly represents a shift in reasoning capabilities or merely fits the structure of these specific problems. Evaluating on a small set of more realistic or semi-structured tasks, which remain computationally manageable, would provide stronger evidence that the proposed supervision paradigm captures reasoning patterns that extend to practical and more complex problem settings. This would meaningfully strengthen the claim without requiring excessive additional resources.

---

### Official Review · Reviewer_BGCK · 2025-10-28

**Soundness:** 3
**Presentation:** 3
**Contribution:** 2
**Rating:** 6
**Confidence:** 4

**Summary:**

The paper proposes TSLM, a training approach that teaches an LLM to internally construct and traverse a tree of reasoning within a single generation, rather than relying on external multi-pass search at inference time. The authors position TSLM as more efficient and easier to train and deploy than external search pipelines, and as a simpler alternative to RL approaches such as GRPO for improving multi-step reasoning and planning. Empirically the paper claims that TSLM improves extrapolation on structured reasoning tasks and helps smaller models approach the performance of larger ones, while also benefiting large models.

The paper situates itself against external search methods, and also cites multi-path generation like self-consistency and recent long-reasoning models. The narrative is consistent with the goal of moving reasoning structure into the model rather than the inference loop. However, there is a relevant gap regarding diversity-driven reasoning frameworks that target breadth before convergence and explicitly reduce redundant reflections with task-agnostic memory.

**Strengths:**

TSLM aims at deployment simplicity. Eliminating external orchestration and multiple model calls is attractive for production where latency and engineering complexity matter. The authors take care to compare conceptually with ToT and with RL, explaining why a supervised approach can be more stable and economical. The stated effect that TSLM helps smaller models close the gap with larger ones is promising for cost-sensitive settings.

The conceptual boundary is easy to communicate. External search explores a tree outside the model, TSLM learns to write the tree inside the generation. That is a clean story, and it could generalize to code or games where branching structure is natural.

**Weaknesses:**

Related work coverage is incomplete on diversity-first inference and redundancy reduction with memory, which is a very proximate thread to the authors’ efficiency and breadth claims. The current submission neither cites nor contrasts with these works (e.g., Lingam et al., ICLR 2025), which could mislead readers about the frontier on exploration efficiency. This needs correction.

Evidence granularity is thin in the visible draft. The claims about improved extrapolation, small-to-large bridging, and efficiency would be more convincing with matched-budget comparisons against ToT, self-consistency, and diversity-driven methods, for example controlling node expansions or total tokens, plus wall-clock latency. The draft does not show whether TSLM outperforms a well tuned inference-time breadth strategy at the same compute.

**Questions:**

(1) Under a fixed budget of model calls and total tokens, how does TSLM compare to ToT or self-consistency, and to a diversity-driven method with a task-agnostic memory bank such as DoT?

(2) What happens if you linearize the tree to a CoT trace with the same token budget, or if you randomize branching order?

**Details Of Ethics Concerns:**

The work is a modeling and training method, uses standard benchmarks, and does not involve sensitive data or human subjects.

---

> ### Author Response · Authors · 2025-11-20
>
> Thank you for the detailed review. We'll address your concerns below, but please first check the general comment which provides important context.
>
> # W1: Related Work Coverage
>
> We first want to clarify an important conceptual distinction between our methods and self-refinement methods like DoT.
> The methods fall into three categories with different goals:
>
> * **Training LMs for offline planning (our focus): SC, PC, GRPO, TSLM** - these aim to train the model to generate search trees for problems presented at inference time, like what reasoning models like o1 or deepseek-R1 are doing.
> Scaffolding for offline planning (which we have for baseline):
> * **Offline Scaffolding (ToT, RAP)** - these apply external search algorithms on top of a fixed base model
>
> * **Self-refinement for online planning: DoT, Reflexion, LATS, xoT** - these methods integrate information from interaction/feedback to iteratively revise reasoning
>
> **Our work targets offline planning capability** , teaching the model to construct good search trees when first seeing a problem. The self-refinement methods you mentioned address online planning - revising solutions based on external feedback or execution results. While these appear similar (both involve exploration), they serve different purposes. For instance, in code generation, offline planning means generating well-structured code initially, while online planning means debugging code based on test failures.
>
>
> Online planning methods could potentially enhance offline capability (by having the model self-revise without external feedback), but it’s more natural to consider this **orthogonal** to our current contribution. We made this clear on our revision, check Appendix A Q4
> and Appendix B!
>
> # W2: Evidence Granularity and Matched-Budget Comparison
>
> As we described in general comment #1, our ToT evaluation is actually **ToT@100** (success if any of first 100 terminal nodes is correct, which is much loosed condition compared to ToT@1 selection from the original ToT paper).
>
>
> The ToT@100 evaluation creates an upper bound for related sampling approaches like ToT@1 (Yao et al.), RAP@1 (Hao et al.), or Self-consistency@100 (Wang et al.). Mathematically, we can establish both  E[P(ToT@1)] ≤ E[P(ToT@b^d)] and E[P(RAP@1)] ≤ E[P(ToT@b^d)] (ToT@b^d passes the instance if one of the whole search tree with beam width b and depth d passes). Also, E[P(Self-consistency@b^d)] ≤ E[P(Pass@b^d)] =E[P(ToT@b^d)], since tree expansion has the same expectation with parallel sampling.
>
> Our emperical number of candidate 100 was found that E[P(ToT@b^d)] $\simeq$ E[P(ToT@100)], which means the first 100 nodes are sufficient for convergence in our setting.
>
>
> This means our ToT baseline already provides an upper bound for what these other scaffolding methods could achieve. Given that TSLM@1 outperforms or matches ToT@100, adding RAP or other selection baselines would not change the core finding.
>
>
> **For the matched-budget comparison you requested:** See general comment #3 and Rebuttal Table 2. TSLM starts better (61.3% vs 52.9% at pass@1), converges faster (TSLM@5 vs ToT@100), and reaches higher final performance (67.2% vs 62.3%). Wall-clock time also favors TSLM due to single forward pass generation versus multiple independent model calls. We also made this point clearer on the revision section 5.3.
>
>
>
>
> # Q1: Fixed Budget Comparison
>
> General comment #3 and Rebuttal Table 2 directly answer this. At any fixed # candidate budget (k), TSLM outperforms ToT. The performance gap is largest at low k (8.4% at pass@1) and persists even as k increases.
>
> For diversity-driven methods with memory banks like DoT, we acknowledge this would be an interesting comparison but consider it somewhat orthogonal given the online vs offline planning distinction noted above.
>
> # Q2: Linearizing Tree and Randomizing Branching Order
>
> Linearizing the tree to a CoT trace with the same token budget is exactly what Sequential Cloning (SC) does. SC takes the correct solution path from the search tree and trains on it as a linear sequence. The results show SC achieves 55.8% on GSM8K versus TSLM's 61.6%, demonstrating the value of tree structure over linearization.
>
>
> We also internally experimented with randomizing the reward-based ordering of branches but didn't include it in the submission because it slightly degraded performance. Specifically, random ordering resulted in lower TSLM@1 accuracy and similar convergence points to the reward-ordered version. Appendix G.2 (Section 6.3 in revision) provides cleaner analysis of tree search strategies (BFS vs DFS) under the reward-ordered setting, which is why we focused the narrative there.

---

### Official Review · Reviewer_fgNv · 2025-11-01

**Soundness:** 2
**Presentation:** 3
**Contribution:** 2
**Rating:** 4
**Confidence:** 2

**Summary:**

LLMs can generate sequential solutions but are unable to systematically explore multiple possibilities. The paper introduce a framework to serialize tree exploration into linear sequences for training to enable tree-structured reasoning. The method achieves better performance than baselines on most tasks.

**Strengths:**

The paper is generally clearly written.

The proposed tree-structured language modeling paradigm is interesting.

**Weaknesses:**

The method is helpful on tasks where systematic exploration is needed, but I think more open-ended and real-world tasks are more important. As indicated in Table 1, the proposed method has no benefit in these scenarios. The bolded number in the GSM8k row should be the ToT one, by the way.

The experiment detailed configuration for the ToT baseline is not mentioned. Since the proposed method requires much higher inference-time compute, it is unclear how much the advantage will diminish	if you let all methods use the same budget (say, the total tokens).

**Questions:**

Does the training data for the TSLM method include the 5 tasks that are also used for evaluation?

Can you also try letting baseline methods train on the same training set, by extracting correct path from the tree. Basically, TSLM receives additional training signal and eliminating that factor will make the experiment results more solid.

---

> ### Author Response · Authors · 2025-11-20
>
> Thank you for the thoughtful review. We'll address each of your concerns below, but please first check the general comment, which resolves most of the issues you raised.
>
>
> # W1: Performance on Open-Ended Tasks
> You noted that TSLM shows no benefit on open-ended tasks like GSM8K. This concern stems from the ToT comparison issue we explain in the general comment. Here is the summary of the corrected GSM8K comparison (see general comment #1 and Rebuttal Table 1):
> * Original submission mistakenly reported ToT at 85% (this was a data recording error - we accidentally used a Qwen 2.5 GRPO result)
>
>
> * Actual ToT performance: 62.3% at ToT@100 (100 attempts per problem)
>
>
> * TSLM performance: 61.6% at pass@1 (single attempt)
>
>
> TSLM achieves nearly the same accuracy as ToT while using 1/100th of the pass@k budget.
>
>
> When comparing at the same # of candidate budget (Rebuttal Table 2 in general comment):
>
> * TSLM@1: **61.3%**
>
>
> * ToT@1: 52.9%
>
>
> * TSLM@5: **67.2%**(converges)
>
>
> * ToT@100: 62.3% (converges)
>
>
> TSLM demonstrates clear benefits on open-ended tasks when compared fairly.
>
>
> Regarding the bold number: You're correct that our original table was misleading. The corrected Rebuttal Table 1 in the general comment separates pass@1 methods (SC, PC, GRPO, TSLM) from pass@100 methods (ToT).
>
>
> # W2: ToT Configuration Details
>
>
> As we discussed in the general comment #1, the detailed ToT configuration is:
> * Search algorithm: Breadth-First Search (BFS)
>
>
> * Base model: SC
>
>
> * Beam width: b=5
>
>
> * Temperature: 0.3
>
>
> * Evaluation: ToT@100 (success if any of first 100 terminal nodes is correct)
>
>
> Also you could check the actual time budget would be much more efficient for TSLM than other methods. Though we acknowledge this should have been explicitly stated in the main paper.
>
>
> # Q1: Training Data Overlap
>
> Yes, the training data includes instances from the same 5 task types used in evaluation, but with completely separate instances in train/test splits. For example, for the Game of 24, we used 1.2K training instances, and 100 test instances with different number combinations.
>
>
> # Q2: Training Baselines on Same Data
>
> There seems to be a misunderstanding about Algorithm 1. The "correct path" you mentioned extracting from the search tree is simply the gold trajectory that already exists in our data. This is exactly what Sequence Cloning (SC) does; It trains on just the correct solution path without seeing any of the failed exploration branches.
>
>
> The key comparison is what additional information each method receives. SC sees only correct paths. PC and TSLM both see the complete search trees (correct and incorrect paths), but differ in how they structure this information: PC serializes the tree into a linear sequence while TSLM preserves the tree structure. The performance differences (SC: 55.8%, PC: 55.9%, TSLM: 61.6% on GSM8K) reflect how effectively each method learns from the exploration structure.

---

### Official Review · Reviewer_cbMw · 2025-11-01

**Soundness:** 3
**Presentation:** 2
**Contribution:** 2
**Rating:** 4
**Confidence:** 3

**Summary:**

This paper introduces Tree-Structured Language Modeling (TSLM), enabling language models to perform tree-structured reasoning natively within a single forward generation process.
The thought branching structures are shown into the token sequence through special markers.
Serialized tree traces that capture both successful and failed reasoning paths are used to train standard transformers.
Experiments are conducted on structured planning tasks and open-ended reasoning tasks.

**Strengths:**

1. Tree structured reasoning is reflected into token sequences and only one model call is required. This is different from multiple independent model calls used by Tree-of-Thought.
2. The token-based serialization approach is practical for achieving tree-structured generation using LLMs.
3. The experiments span both structured and open-ended reasoning tasks.

**Weaknesses:**

1. The time cost comparison with methods such as Tree-of-Thought is not presented.
2. As shown in Table 1, the performance of TSLM is not as good as ToT in GSM8K.
3. The token-based tree serialization relies on search trees gotten from other reasoning methods. As such, the performance is also limited by the other reasoning methods.
4. There are some more advanced reasoning methods such as Graph-of-Thought and Everything of Thoughts.

**Questions:**

What is the time cost of the proposed method?

---

> ### Author Response · Authors · 2025-11-20
>
> Thank you for the thoughtful review. We'll address each of your concerns below, but please first check the general comment, which resolves most of the issues you raised.
>
>
>
>
> # W2: Performance Comparison with ToT
>
>
> The key point from the general comment is that **ToT in our experiments is a pass@100 evaluation** (the model succeeds if any of 100 attempts is correct), while TSLM and other baselines are **pass@1** (see general comment #1). This means direct comparison is not apples-to-apples.
>
>
>
>
> Even with this caveat, the results show that:
>
>
>
>
> * TSLM substantially outperforms ToT on search-intensive tasks(e.g., Game of 24: 100% vs 32%, even with ToT getting 100 attempts)
>
>
>
>
> * TSLM achieves comparable performance to ToT pass@100 with just a single attempt on open-ended tasks (see general comment #2)
>
>
>
>
> # W1: Time Cost and Fair Comparison
>
>
> What about fair comparison at the same # of candidates? This is exactly what we show in general comment #3 and Rebuttal Table 2. TSLM not only starts better, but converges higher and faster than baselines.
>
>
>
>
> For time cost comparison:
>
>
>
>
> * vs ToT: TSLM is significantly faster at inference due to causal generation of the search tree in a single forward pass, versus ToT's 100+ separate model calls
>
>
>
>
> * vs Procedure Cloning (PC): TSLM reconstructs the search tree more efficiently by ignoring key-value cache of irrelevant subtrees during inference, while PC must process the entire sequential trace
>
>
>
>
> # W3: Graph-of-Thought and Everything-of-Thoughts
>
>
> We appreciate the mention of these advanced methods, but would like to clarify why we don't consider them the primary comparison point. **Our core contribution is teaching language models to causally reconstruct search trees** within the generation process itself. The comparison with ToT was specifically chosen to demonstrate the inefficiency of external scaffolding methods, which suffer from redundant parallel sampling.
>
>
>
>
> For example, when using parallel sampling methods (scaffolding methods, including GoT and xoT), if we generate two parallel children A and B, we cannot determine whether they represent a meaningful exploration of the search tree or are merely redundant attempts. Causal generation of the search tree is the main principle of our work.
>
>
>
>
> This is why we focused comparisons on PC and GRPO, which also generate search trees causally without parallel sampling intervention. Both GoT and xoT rely on parallel sampling for different candidates, which fundamentally differs from our causal generation approach. (Also wrote in Appendix A Q4 in the revised version)
>
>
>
>
> # W4: Token-Based Tree Serialization and Training Data
>
>
> There seems to be a misconception about our data collection method. Our bootstrapping procedure (Algorithm 1) is specifically designed to handle cases where the base model cannot reliably solve problems. Here we guarantee at least one valid solution in every training tree (line 281), then fill the remaining branches with plausible alternatives from the model. This means even when base model success rate is low (as in Game of 24 or Gridworld), we can construct rich search trees that teach systematic exploration.

---

### Author Response · Authors · 2025-11-20
**Official Clarification on Tree-of-Thought (ToT) Baseline Configuration on GSM8K**

Thank you all for the thoughtful feedback!  We want to start by acknowledging and correcting a critical configuration issue from our September submission that may have caused confusion about our contributions. We revised the paper upon all your concerns, so feel free to check the new version!


# ⚠️ The Core Misalignment: Pass@100(Tree-of-Thought) vs Pass@1(Others) ⚠️


_TL;DR: We made a fundamental mistake in marking and explaining with ToT comparisons. ToT in our experiments is a pass@100 method, while all other methods (SC, PC, GRPO, TSLM) are pass@1. This is like comparing apples to oranges!_


## 1. Tree-of-Thought(ToT) configuration (Missing in the original paper)


Our ToT baseline uses the following configuration across all experiments:


* Search algorithm: Breadth-First Search (BFS)


* Base model for the rollout: model SFT-ed by the standard sequence format (SC in our paper)


* Sample ordering: RAP (Hao et al., Prioritize searching higher process reward first)


* Beam width: b=5


* Temperature: 0.3


* Evaluation: ToT@100 (we examine the first 100 terminal nodes and mark the instance as correct if any candidate reaches the correct answer)


*We set ToT@k=100 in our setting, because k=100 already gave a good convergence point within the search tree.


One misconception among reviewers is that our method is not performing well compared to ToT. The truth is, however, ToT implementation in our paper is a test-time scaling method where we give multiple shots for the model to solve the same problem with parallel inference. Unlike other methods (including ours), they get instances right if at least one of 100 attempts is right.


I am deeply sorry about the confusion and we should have made a more understandable figure for ToT in Table 1.


## 2. Rebuttal Table 1 (Updated from original Table 1)


What #1 means is that the original Table 1 should have looked more like this:


| Tasks | SC | PC | GRPO | TSLM (ours) | | ToT@k (test time scaling from SC) |
|---------:|:--------:|----------:|----------:|----------:|:--:|----------|
| # candidates(&darr;) | 1 | 1 | 1 | 1 | ║ | 100 |
|   Game of 24    |  17.0%  | 47.0%     | 15.0% | **100%**| ║ | 32.0% ; _changed from 17%_|
|   Gridworld (10x10)    |  78.2%   | 99.7%     | 24.0% | **100%**| ║ | 95.0%|
|   Gridworld (20x20); _we reevaluated with the same model_ |  33.0%   | 81.1%     | 6.0% | **91.5%**| ║ | 42.7%|
|   ProntoQA    |  99.7%   | 97.5%     | 99.8%| **100%**| ║ | 100%|
|   GSM8K    |  55.8%   | 55.9%     | 60.8%|**61.6%**| ║ | 62.3% ; _changed from 85%_|


*For your recall,
* SC (Sequence Cloning) is a standard CoT SFT method.
* PC (Procedure Cloning) and GRPO (Group Relative Policy Optimization) are each SFT and RL methods to reproduce o1-like search tree in one model call.
* TSLM (Tree-Structured Language Modeling) also reproduces o1-like search tree, but makes sure the model doesn’t attend to irrelevant subtree during inferencing each node
* ToT@k (Tree-of-Thought) is a tree-like scaffold on top of SC to reproduce the search tree via multiple model calls.


### So What Changed?


* ToT score correction: We accidentally swapped some of the ToT results with other results by mistake. For Game of 24, it should be 32.0% instead of 17.0% (is a standard SFT result), and for GSM8K, it should be 62.3% instead of 85.0% (is a GRPO result for Qwen 2.5 7B, see #4).  Also we reported some of the Gridworld 20x20 performance with Qwen2.5 0.5B model, so we also changed it with our base llama 3 8B model performance (no big change in narration though)


* I am sorry for the mistake, and it’s totally natural to doubt that we are manipulating on GSM8K results. However, if you see the increment in Figure 3  (Figure 4 of the original version), you would see that 85% doesn’t make any sense. The purple line (SC) has already converged to near 62% at $k=9$. ToT is a convergence point of the graph, and there’s no way it could be 85%.


* The real comparison: Even in GSM8K, when you give ToT 100 chances to find the answer, it only barely beats TSLM's **single** attempt (62.3% vs 61.6%).

---

> ### Author Response · Authors · 2025-11-20
> **General comment cont.**
>
> ## 3. The Fair Comparison: Test-Time Scaling (Rehighlighting Appendix G as Section 5.3)
>
>
> _TL;DR: If we do the fair test-scaling comparison of which ToT did, TSLM starts better, converges faster, and also takes lower budget!_
>
> So the immediate follow-up question would be: What happens if we have a "fair comparison" across methods that can be scaled during inference? We compared these two types of scaling against ToT:
>  * TSLM: Controls the number of candidates $k$ by verifying the first $k$ terminal states from Breadth-First Search (BFS) of the generated tree structure
> * Procedure cloning (PC): Verifies the first $k$ terminal states from the sequential search trace
>
>
> The spirit under this is, even though TSLM and PC can verify each leaf node being correct or not, we ignore the verification and see how their performance increases when we collect all the candidates they produce, similar to ToT.
>
>
> Here's what happens when we compare methods at the same pass@k budget:
> | Test-time scaling (# candidates)   | 1 | 2  | 3 | 4 | 5 | 6| 7| 8| 9| 10| 100(convergence point) |
> |---------:|:--------:|:--------:|----------:|----------:|----------:|:--:|----------:|---------:|:--------:|:--------:|----------:|
> |   ToT   | 52.9% | 56.8%  | 59.0% |60.2% | 60.7% | 61.2% | 61.5%| 61.5%| 61.7%| 61.9%| 62.3%|
> |   PC   | 53.3%|  58.9% | 61.9% | 62.7% | 63.1% | 63.3%| 63.5% | 63.5%| 63.5%| 63.6%| 63.6% |
> |   TSLM   | **61.3%** | **66.3%**  | **67.0%** | **67.2%** | **67.2%** | **67.2%**| **67.2%**| **67.2%**| **67.2%**| **67.2%**| **67.2%** |
>
>
> We also plot the actual wallclock time(=time budget) that each methods require
> | Avg wallclock time over candidates(sec, &darr;)  | 1 | 2  | 3 | 4 | 5 | 6| 7| 8| 9| 10| min(100, convergence point) |
> |---------:|:--------:|:--------:|----------:|----------:|----------:|:--:|----------:|---------:|:--------:|:--------:|----------:|
> |   ToT   | 3.76 | 5.40 | 7.21 | 8.76 |10.55 | 12.34 | 14.56 | 16.48| 18.42| 20.23 | 186|
> |   PC   | 2.29|  3.38 | 4.62 | 5.91 | 7.46 | 8.80 | 9.93 | 10.96 | 12.79 | 13.47 | 14.68 |
> |   TSLM   | **2.07** | **2.95**  | **4.23** | **5.70** | **6.86** | **8.05**| **9.52**| **10.15**| **10.49**| **10.53**| **10.53** |
>
>
> *all experiments are done with VLLM version 0.6.6.
>
>
> The time explosion of ToT is due to redundant sampling from parallel sampling, while PC and TSLM are reconstructing the search tree efficiently with causality. TSLM is slightly faster than PC due to higher computation efficiency by ignoring irrelevant subtree during inference.
>
>
> Sorry for underemphasizing this section in the September submission, which might have questioned reviewers about the fair comparison with test-time scaling method and its effectiveness.
>
>
> ## 4. Bonus: Results on Different Models (Extension of Figure 6)
>
>
> We also have results on Qwen 2.5 that we didn't fully highlight:
>
>
> | GSM8K result | Base model | SC | PC | GRPO | TSLM (ours)
> |---------:|:--------:|:--------:|----------:|----------:|----------:|
> |   Llama 3 8B   | 51.4% |  55.8%   | 55.9%     | 60.8%|**61.6%**|
> |   Llama 3 8B Instruct (exp on further SFT-ed model)  | 51.4% |  60.5%   | 56.4%     | 62.7%| **62.9%**|
> |   Qwen 2.5 7B  (exp on a more modern model)  |  59.8%   | 81.9%   |82.0%  | 85.0% |**87.4%**|
>
>
> We'll address each reviewer's specific concerns in separate threads below. We hope this clears up the confusion about our experimental setup! Happy to discuss any of these points further.
>
> **Once again, we revised our paper upon these fixes/addition so feel free to check it out!!**

---

### Meta-Review · Area_Chair_eTWh · 2025-12-29

**Summary:**

While the original reviews were mixed between weak accept and weak reject, the primary concern of the reviewers regarding the baseline was addressed in the rebuttal. Other concerns regarding related work were also addressed. Eventually, the paper addresses an important problem ("Divergent Thinking") by using a toy environment, which is the main weakness. I recommend a weak accept at this point.

**Reviewer Concerns:**

no concerens

**Reviewer Scores:**

not relevant

---

### Decision · Program_Chairs · 2026-01-26

Accept (Poster)